ecology, evolution

host traits, arthropod, DNA metabarcoding, wood-decay fungi, fruit body, insect–fungus interactions

**Authors for correspondence:**
Lisa Fagerli Lunde
e-mail: lisa.fagerli@nmbu.no
Sundy Maurice
e-mail: sundymaurice@gmail.com

# DNA metabarcoding reveals host-specific communities of arthropods residing in fungal fruit bodies

Lisa Fagerli Lunde[1], Tone Birkemoe[1], Håvard Kauserud[2], Lynne Boddy[3], Rannveig M. Jacobsen[1,4], Luis Morgado[2], Anne Sverdrup-Thygeson[1] and Sundy Maurice[2]

[1]Faculty of Environmental Sciences and Natural Resource Management, Norwegian University of Life Sciences, Ås, Norway
[2]Section for Genetics and Evolutionary Biology, University of Oslo, Blindernveien 31, 0316 Oslo, Norway
[3]Cardiff School of Biosciences, Cardiff, UK
[4]The Norwegian Institute for Nature Research, Oslo, Norway

LFL, 0000-0002-4250-8182; HK, 0000-0003-2780-6090; LB, 0000-0003-1845-6738; RMJ, 0000-0003-4055-1096; AS-T, 0000-0002-3122-2250; SM, 0000-0002-5376-0981

Biological communities within living organisms are structured by their host's traits. How host traits affect biodiversity and community composition is poorly explored for some associations, such as arthropods within fungal fruit bodies. Using DNA metabarcoding, we characterized the arthropod communities in living fruit bodies of 11 wood-decay fungi from boreal forests and investigated how they were affected by different fungal traits. Arthropod diversity was higher in fruit bodies with a larger surface area-to-volume ratio, suggesting that colonization is crucial to maintain arthropod populations. Diversity was not higher in long-lived fruit bodies, most likely because these fungi invest in physical or chemical defences against arthropods. Arthropod community composition was structured by all measured host traits, namely fruit body size, thickness, surface area, morphology and toughness. Notably, we identified a community gradient where soft and short-lived fruit bodies harboured more true flies, while tougher and long-lived fruit bodies had more oribatid mites and beetles, which might reflect different development times of the arthropods. Ultimately, close to 75% of the arthropods were specific to one or two fungal hosts. Besides revealing surprisingly diverse and host-specific arthropod communities within fungal fruit bodies, our study provided insight into how host traits structure communities.

## 1. Introduction

Biological communities in or on living organisms are structured by their host's traits, such as size or persistence, which can influence both the survival and biotic interactions of species in the community via selection processes [1–3]. The relative importance of these traits varies with spatial scale and dispersal rates of the species living in close association with their host [4,5]. Nevertheless, a better understanding of how host traits affect biodiversity and community composition is needed, in particular from poorly explored habitats. Arthropods are ubiquitous in almost every ecosystem worldwide and, because of their small sizes, many arthropods develop in habitats that comprise a single food item (e.g. [6,7]). As these items represent discrete habitat patches, they are ideal for studying how host traits relate to community composition and species richness. Physical traits like host size, shape and toughness can have various effects on arthropod communities, as has been thoroughly studied for plants and their herbivores [8–10]. Host persistence is also an important determinant

of species' life cycles and dispersal abilities. Short-lived hosts select for strong dispersers with fast developmental times, for example, the beetles *Diaperis boleti* and *Tetratoma fungorum* breeding in fungal fruit bodies [11,12]. While, in the long-lived habitat of hollow trees, *Osmoderma eremita* develops slowly and has limited dispersal abilities [13,14]. In the present study, we explore the importance of host traits for diversity and community composition in a poorly explored interaction network: arthropods inside living fruit bodies of fungi.

The interactions between arthropods and fungi in dead wood present an intriguing study system. Wood-decay fungi can decompose the recalcitrant lignocellulose within dead wood to render more palatable and nutritious food sources for arthropods [15,16]. Their fruit bodies provide nutrition and shelter to a large community of arthropods—mainly insect larvae, springtails, mites and spiders—which probably colonize the habitat via spore tubes or crevices in the bark where the fruit body is attached [17,18]. Certain fruit body traits, like size, morphology, toughness and persistence vary between fungal species. For example, some wood-decay fungi can produce voluminous fruit bodies that protrude from the wood like shelves (*pileate*), while others stretch like thin blankets along with the log and have large spore-producing surface areas (*resupinate*) [19]. Fruit body persistence is highly variable between wood-decay fungi, lasting between a few weeks for some species and several years for others, and it correlates with toughness—short-lived fruit bodies are soft while long-lived are progressively tougher [20,21]. Persistence can determine host specificity of arthropods, as less predictable resources, for example, short-lived fruit bodies, move towards generalization in a community [22,23]. There are indeed indications that persistent fruit bodies host more specialized arthropods [22,24,25].

Most arthropods associated with fruit bodies of wood-decay fungi spend their immature life inside their host as fungivores, detritivores, predators or parasitoids [26]. Yet, there is considerable variation in the behaviour and life-history strategies among these arthropods, for instance in terms of dispersal ability. While most adult insects can fly between fruit bodies, mites and spiders disperse with the wind, by walking or hitchhiking with other animals (*phoresy*). Development times of arthropods are often limited by the persistence of the fungal host, although the duration of the life cycle is somewhat fixed between different arthropod orders. True flies are the dominant arthropods present in ephemeral agaric mushrooms [27–29] and, together with some orders of minute mites, have exceptionally short life cycles lasting a few weeks or even days [30]. Oribatid mites, lepidopterans and beetles, on the other hand, in general develop more slowly and are found in more persistent fruit bodies [30–34]. There is a succession of arthropods within the fruit bodies as they age and decay, with a higher diversity in dead fruit bodies [17,35] where several species, e.g. most ciid beetles [21,26], are exclusively found. Although the fauna is seemingly different in living fruit bodies than in dead [35,36], there is little data on the arthropod communities that colonize living fruit bodies. Furthermore, many studies have looked at how fruit body traits affect different arthropod communities (e.g. [37,38]), but usually within one taxonomic group or trophic guild at a time. The studies that have considered all arthropods are either exploratory, i.e. not inferring community patterns based on host traits [17,26,36], or limited to one or two fungal hosts [25,35,39]. However, to get a solid understanding of the effects of host traits in the fungi-arthropod system, we need to assess arthropod communities across different hosts.

In this study, we compared the diversity and community composition of arthropods in living fruit bodies of 11 species of wood-decay fungi. We tested the extent to which certain fruit body traits, namely size, thickness, surface area, morphology, persistence and toughness, were important. We amplified the mitochondrial COI marker and applied DNA metabarcoding to identify arthropods, and addressed the following questions: (Q1) *Does the diversity of arthropods depend on fruit body traits?* We expected diversity to increase with fruit body size, because bigger fruit bodies have larger spatial heterogeneity and potentially attract more arthropod species. We expected diversity to increase with fruit body persistence because the habitat is more stable, and there is more time to accumulate species. As for morphology, we expected arthropod diversity to be higher in pileate fruit bodies because larger volumes typically host more niches, e.g. sections within the fruit body that decompose at different rates. (Q2) *Are arthropod communities shaped by fruit body traits?* We expected distinct communities in fungi with different size, morphology, persistence and toughness. Further, we expected short-lived fruit bodies to host more arthropods with short development times, and *vice versa*. (Q3) *Are there species-specific co-occurrences between arthropods and fungal hosts?* We expected a relatively high degree of host specificity and that toughness might relate to arthropod host selection.

## 2. Material and methods

### (a) Sampling and compilation of metadata

We studied 11 wood-decay fungal species from an old-growth boreal spruce-dominated forest in Southeastern Finland (Issakka, Kuhmo). We targeted both species with common and rare occurrences on dead spruce in Fennoscandia, including 10 polypore fungi, namely *Amylocystis lapponica* (amylap), *Antrodia serialis* (antser), *Gloeophyllum sepiarium* (glosep), *Fomitopsis* (*Rhodofomes*) *rosea* (fomros), *Fomitopsis pinicola* (fompin), *Phellopilus nigrolimitatus* (phenig), *Phellinidium ferrugineofuscus* (phefer), *Phellinus* (*Fuscoporia*) *viticola* (phevit), *Postia cyanescens* in *Postia caesia* complex, *Trichaptum abietinum* (triabi) and the corticoid species *Phlebia centrifuga* (phecen) (electronic supplementary material, table S1). In October 2016, we collected between 19 and 26 individual fruit bodies from distinct spruce logs for each fungus species. The study area and sampling scheme have been described in detail in previous publications [40,41].

For each of the 11 fungal species, we compiled information on six fruit body traits that potentially impact arthropod communities (electronic supplementary material, table S1) [19,42,43]. These were the *fruit body size*, mean *fruit body thickness* (cm), *mean hymenophore surface area* ($cm^2$), *morphology* (resupinate/pileate), *persistence* (short- and long-lived; corresponds to annual and perennial, respectively) and *hyphal system complexity* (monomitic/dimitic/trimitic). The latter describes fruit body toughness where species with monomitic hyphae have softer fruit bodies, and species with dimitic and trimitic hyphal systems are progressively tougher [20,21]. Fruit body persistence and toughness are highly correlated fruit body traits. All short-lived species have monomitic hyphal systems, with the exception of the dimitic *Trichaptum abietinum*. We therefore included the variable 'persistence' only in the univariate analyses.

## (b) DNA extraction and metabarcoding

Details of sample processing and DNA extraction were provided in Maurice, Arnault [40]. Briefly, we processed all fruit bodies similarly, by cutting out the outer surface layer, to avoid surface contaminants. We cut off between 10 and 15 small pieces of approximately 5 mm$^2$ from the subiculum layer of living fruit bodies and ground them in 800 µl of 2% CTAB and 1% beta-mercaptoethanol using a Retsch MM200 mixer ($4 \times 45$ s at 25 oscillations). We extracted DNA using a modified CTAB extraction protocol [44] and cleaned the DNA extract with the E.Z.N.A Soil DNA kit (Omega Bio Tek) starting with the addition of the HTR reagent and following the manufacturer's protocol. DNA was eluted in 100 µl elution buffer, quantified with Qubit ds DNA BR Assay kit (Life Technologies) and standardized with 10 mM Tris to a concentration range of 5–10 ng µl$^{-1}$ total genomic DNA.

We amplified part of the COI gene using the primer pair BF3/BR2 [45], resulting in an amplicon of approximately 421 bp. We designed a combination of 96 twin-tagged primer pairs with a unique sequence of seven nucleotides at the 5′ end in addition to one or two nucleotides as heterogeneity spacer (hereafter *linker tag*). The 25 µl PCR mix consisted 2.5 µl $10 \times$ Gold buffer, 0.2 µl dNTP's (25 nM), 1 µl reverse and forward primers (10 µM), 2.5 µl MgCl2 (50 mM), 1 µl BSA (20 mg ml$^{-1}$), 0.2 µl AmpliTaq Gold polymerase (5 U µl$^{-1}$, Applied Biosystems, Thermo Fisher Scientific) and 1.5 µl of DNA template. DNA amplification consisted of an initial phase of denaturation and Taq-activation for 5 min at 95°C, followed by 30 cycles of 95°C for 30 s, annealing phase at 46°C for 30 s and extension phase at 72°C for 50 s, and a final extension phase at 72°C for 5 min. We amplified 288 samples including 13 technical replicates, three negative (water) and three positive PCR controls, with 1 µl of DNA extracted from *Ctenolepisma longicaudata* (Zygentoma). All amplicons were checked on a 1% agarose gel before normalization using the SequalPrep Normalization Plate Kit (Invitrogen, Thermo Fisher Scientific, Waltham, MA, USA) and eluted in 20 µl elution buffer.

The 96 uniquely barcoded amplicons within each of the three libraries were pooled, concentrated and purified using Agencourt AMPure XP magnetic beads (Beckman Coulter, CA, USA). The quantity and quality of the DNA were checked using Qubit (Invitrogen, Thermo Fisher Scientific, Waltham, MA, USA). The three libraries were barcoded with Illumina adapters, spiked with 20% PhiX and pooled to equimolar concentrations and sequenced on an Illumina MiSeq lane (Illumina, San Diego, CA, USA) using V3 chemistry and a $2 \times$ 300 bp paired-end at StarSEQ (Mainz, Germany). The resulting metabarcoding dataset comprised 26 705 967 sequences.

## (c) Bioinformatics analyses

Prior to the analyses, the linker tag between the barcode and Illumina adapter was removed after testing the effect of not removing it (electronic supplementary material S3.1). The raw forward and reverse sequences were demultiplexed independently on a sample basis using CUTADAPT v. 2.7 [46], allowing 26 bp overlap between barcode tags and sequence primer, without indels and discarding sequences shorter than 100 bp. DADA2 v. 1.14 [47] was used to: filter low-quality sequences (max. error 2 and overlap between primer pairs), dereplicate, correct read errors based on a machine learning model built from the sequence data, merge forward and reverse sequences (min. overlap 12 bp) and remove two-parent chimeras *de novo*. This resulted in 259 samples and 8 913 293 sequences which were used to build an amplicon sequence variants (ASV) table. Clustering was performed at 97% similarity with VSEARCH [48]. Taxonomy was assigned with BLAST+ v. 2.8.1 against the BOLD and NCBI databases. We used the LULU

algorithm [49] with default settings to correct for potential over-splitting of OTUs. We tested for the effect of not clustering the ASVs (electronic supplementary material S3.2), but retained the more conservative approach, i.e. the clustered dataset, to avoid overestimation of biodiversity [41]. After checking and excluding controls and replicates (see electronic supplementary material S3.2), the dataset contained 8 504 469 sequences and 8891 OTUs from 244 retained samples. Of these, 6 600 905 sequences and 3606 OTUs had annotations. We retained only OTUs annotated to Arthropoda (46% of annotated OTUs), which resulted in a dataset of 3 270 948 sequences and 1664 OTUs. The remaining OTUs were annotated to Ochrophyta (7.2%), Vertebrata (5%), Basidiomycota (4.5%), Mollusca (4.2%) and Ascomycota (3.9%), among others.

## (d) Statistical analyses

All data exploration and analyses were done with R v. 4.0.3 [50] and figures were visualized with the package GGPLOT2 [51]. Rarefaction and alpha diversity analyses were performed with the package PHYLOSEQ [52]. Prior to the analyses, the arthropod-only dataset was filtered by removing OTUs with less than 10 sequences, resulting in 1234 OTUs. We then rarefied at a subsampling depth corresponding to the 25th percentile (i.e. 1416 sequences per sample) removing 61 samples and 96 OTUs. The subsampling depth was chosen after the evaluation of the rarefaction curves per host species (electronic supplementary material, figure S5). Testing of alpha diversity (Q1) was performed with linear mixed models (LMM) from package LME4 [53] and with Shannon diversity (H′) as a response variable. Six fruit body traits (size, mean thickness, mean surface area, morphology, persistence and hyphal system complexity) were inferred as explanatory variables and tested as single-covariate models with host species as a random effect and restricted maximum likelihood as the optimization criterion. To evaluate which variable had the strongest effect, we compared the Akaike's information criterion (AIC) [54], marginal $R^2$ and significance ($\alpha = 0.05$) of the five models (electronic supplementary material S8). To explore the main gradients of the rarefied arthropod composition (electronic supplementary material S6), we used global non-metric multidimensional scaling (NMDS) [55] in package VEGAN v. 2.5–7 [56]. We removed three outlier samples corresponding to the species *F. pinicola* and calculated Bray–Curtis dissimilarities (details in the electronic supplementary material S6). The axes were sorted by most variation explained and scaled in (half-change) units of compositional turnover [57]. The relationships between arthropod composition and all variables (i.e. beta diversity; Q2) on fruit body characteristics were assessed in three ways. First, they were fitted onto the NMDS by the function envfit (from VEGAN) based on factor averages and vectors for continuous variables, and the significance of their fit was tested. Second, the non-parametric, permutational multivariate analysis of variance (PERMANOVA) test with 999 permutations on Bray–Curtis dissimilarities, following a forward model selection to compare the models with an adjusted AIC based on the residual sum of squares from the PERMANOVA models [58]. Finally, we employed a forward model selection with partially constrained ordinations (CCA) [59] of each explanatory variable based on Monte Carlo permutation tests (999 perm., $F$ as a criterion). We explored co-occurrence patterns between arthropod OTUs and host fungi with a network-based analysis (Q3): significant co-occurrences between the rarefied arthropod dataset and fungal hosts were identified by a multi-level pattern analysis integrated into the package INDICSPECIES [60]. This analysis calculates the indicator value of each OTU to combinations of all host fungi and tests the statistical significance of the strongest co-occurrences after 999 permutations [61,62]. To account for multiple comparisons, we

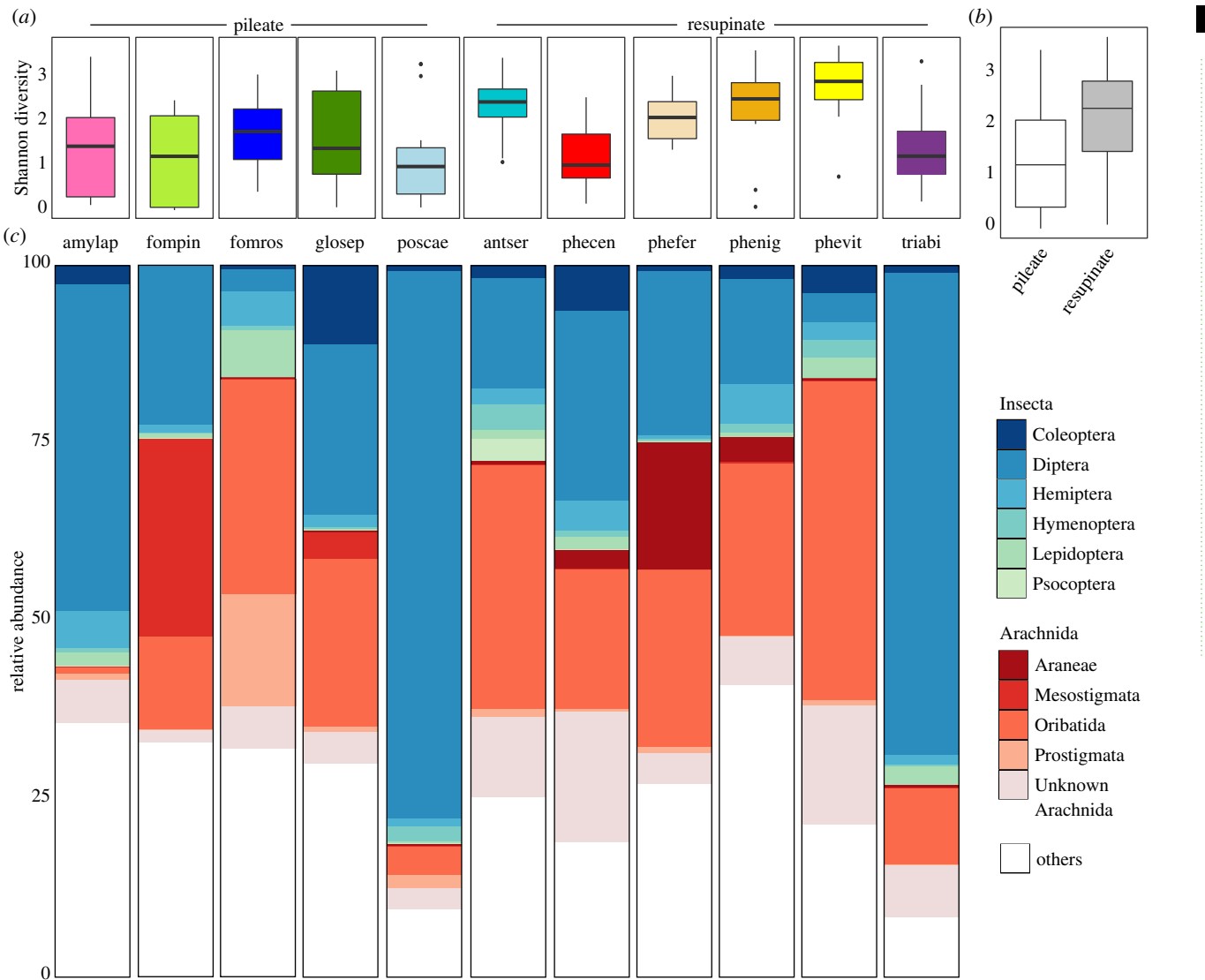

**Figure 1.** Alpha diversity and taxonomic composition of arthropod OTUs (rarefied at 1416 sequences per sample) amplified from 183 fungal samples. (*a*) Box plots showing variation in Shannon diversity of arthropods between the 11 species of fungal hosts. (*b*) Box plots showing estimated Shannon diversity of arthropods in pileate and resupinate fungal fruit bodies. The levels were significantly different when fitted in an LMM with fungal host as random effect. (*c*) Mean percentage of arthropod orders between fungal hosts. The 10 most common orders are coloured: class Arachnida in red and class Insecta in blue and green.

used Benjamini–Hochberg adjusted *p*-values [63] with $\alpha = 0.05$. We visualized the co-occurrence relationship with package IGRAPH [64] as a tripartite network and manually adjusted the layout based on a Sugiyama layout algorithm [65].

## 3. Results

### (a) Arthropod alpha diversity and taxonomic composition

We investigated the diversity and taxonomic distribution of the 1138 arthropod OTUs identified by DNA metabarcoding from living fruit bodies of 11 species of wood-decay fungi. Median Shannon diversity ranged from H′ = 2.92 in *Phellinus viticola* to H′ = 0.99 in *Postia caesia* (figure 1*a*). When investigating how fruit body characteristics explained variation in arthropod diversity, the single-covariate model including fruit body morphology (pileate/resupinate) was the only one deemed better than the null model (ΔAIC = 1.9; $R^2 = 0.11$, $p = 0.046$) (tables 1 and 2; electronic supplementary material, table S8.1; figure 1*b*). Three pileate species,

**Table 1.** Comparing five single-covariate LMMs fitting fungal fruit body traits to Shannon diversity of arthropods (fungal host is random effect).

| explanatory variable | AIC value | degrees of freedom | marginal $R^2$ |
|---|---|---|---|
| morphology | 495.53 | 4 | 0.110 |
| null model | 497.43 | 3 | 0 |
| size | 498.76 | 5 | 0.063 |
| hyphal system | 500.23 | 5 | 0.023 |
| hymenophore area | 500.30 | 4 | 0.043 |
| thickness (mean) | 504.10 | 4 | 0.053 |

*Amylocystis lapponica*, *Fomitopsis pinicola* and *Gloeophyllum sepiarium*, varied widely in diversity between samples (figure 1*a*).

**Table 2.** Single-covariate LMM output showing Shannon diversity of arthropods explained by fruit body morphology with fungal host as random effect. Maximum-likelihood is optimization criterion and *p*-values are calculated from Satterthwaite's approximation with $\alpha = 0.05$ (significant value in italics). Resupinate fruit bodies as reference level (intercept). s.e. = standard error, d.f. = degrees of freedom.

|  | estimate | s.e. | d.f. | *t*-value | *p*-value |
|---|---|---|---|---|---|
| (intercept) | 2.0974 | 0.1991 |  |  |  |
| *Morphology* (*pileate*) | −0.6856 | 0.2941 | 8.6295 | −2.331 | *0.0458* |

Class Insecta and Arachnida represented 43.7% and 33.8% of the arthropod sequences, respectively. Around one fifth of the sequences (21.2%) were not annotated to class-level (unknown arthropods), ranging from 7.1% in *Trichaptum abietinum* to 37.7% in *Phellinidium ferrugineofuscus*. Also, 38.3% of arachnids were not annotated to any order (unknown Arachnida). Diptera, then Oribatida, were most common among the annotated orders of arthropods, followed by Coleoptera, Mesostigmata, Hemiptera, Lepidoptera, Psocoptera, Hymenoptera, Prostigmata and Araneae (figure 1*c*). The most common families were in descending order: Bolitophilidae (Diptera), Chironomidae (Diptera), Oppidae (Oribatida), Cecidomyiidae (Diptera) and Mycetophilidae (Diptera). Short-lived fungi—with the exception of the corticoid *Phlebia centrifuga*—had very high proportions of insects, mostly true flies: *P. caesia* contained 77% true flies, *T. abietinum* 68% and *A. lapponica* 46% (figure 1*c*).

## (b) Community composition

The different fungal species hosted distinct arthropod communities, a pattern that was very pronounced for *P. caesia* and the three species with trimitic fruit bodies: *G. sepiarium*, *Fomitopsis rosea* and *F. pinicola* (figure 2). From the three multivariate analyses, envfit (table 3), PERMANOVA (electronic supplementary material, table S9.1) and constrained ordination (electronic supplementary material, table S9.2), all variables explained the arthropod community composition. The main gradient of the NMDS was related to hyphal system complexity in the fruit body, i.e. toughness, and the proportion of different arthropod orders. More precisely, the gradient went from a higher proportion of true flies and monomitic hyphal system in the fruit bodies, to a higher proportion of beetles, oribatid mites and trimitic hyphal system. The second gradient of the NMDS was related to hymenophore area and the proportion of oribatid mites, but inversely related to fruit body thickness and the proportion of spiders (table 3).

## (c) Indicator species analysis and co-occurrence patterns

Using the results obtained from an indicator species analysis, we created a network of 117 arthropod OTUs showing significant associations with one or more fungal hosts (figure 3*a*; electronic supplementary material, table S10). Of these, 54 OTUs were specific to one host (46.2%), 29 were associated with two hosts (24.8%) and the remaining 34 (29%) were associated with between three and 10 hosts. As much as 60.5% of all insects (23 OTUs) were specific to only one fungal host, representing true flies, beetles, lepidopterans, hymenopterans and true bugs. The corticoid *P. centrifuga* was associated with nearly half of all arthropods

in the network (49 OTUs), although with only three specific associations. Excluding species-specific co-occurrences, only 14 arthropods were specific to a hyphal system—and 10 of these co-occurred in monomitic fruit bodies (figure 3*b*). However, 77 arthropods co-occurred in mono- or dimitic, but not trimitic, fruit bodies. Twenty arthropods did not seem to have any preference for a hyphal system (16 co-occurring in all; four in mono- or tri-, but not dimitic, fruit bodies).

## 4. Discussion

### (a) Does the diversity of arthropods depend on fruit body traits?

Arthropod diversity was driven by fruit body morphology, and surprisingly, we identified a higher diversity in resupinate than pileate fruit bodies. By contrast, a study on 58 wood-decay fungi found that ciid beetle richness was highest in pileate fruit bodies [38]. Wood-decay fruit bodies, especially pileate, harbour several niches as they consist of different structural layers that vary in texture [17], and often within the same fruit body there are sections that vary in lifespan and microclimatic conditions [18]. The main finding from Thorn, Müller [38] is in line with the hypothesis on 'habitat heterogeneity', stating that structural complexity in a habitat increases species diversity [66]. This study, however, focused on a single family of obligatory fungivores—the ciid beetles—which primarily inhabits dead fruit bodies [38]. While, here, we investigated the *whole* arthropod community in living fruit bodies, which can differ for several reasons. First, the relative importance of spatial heterogeneity can vary during succession, as has been shown for insects [67,68]. Second, arthropod taxa might respond differently to spatial heterogeneity, due to divergence in e.g. home range and habitat requirements between taxa [66,69]. In our study, however, resupinate fruit bodies positively affected the diversity of all arthropods, which leads us to ask why. A plausible explanation is that the large attachment and spore-producing surface areas of resupinate fruit bodies facilitate colonization by arthropods. The fruit bodies of wood-decay fungi are seemingly hard to penetrate and arthropods enter them often through pores in the spore-producing layer or between furrowed bark where the fruit body is attached [17,18]. Moreover, traditional island biogeography theory states that when the number of individuals in an environment is small, the species depends on frequent colonization to avoid local extinction [70]. This might be true for arthropods in living fungal fruit bodies as they usually seem to harbour few individuals [35,36], a common phenomenon in short-lived habitats, in general [71]. Although resupinate fruit bodies may host a larger diversity of arthropods because they allow more entry

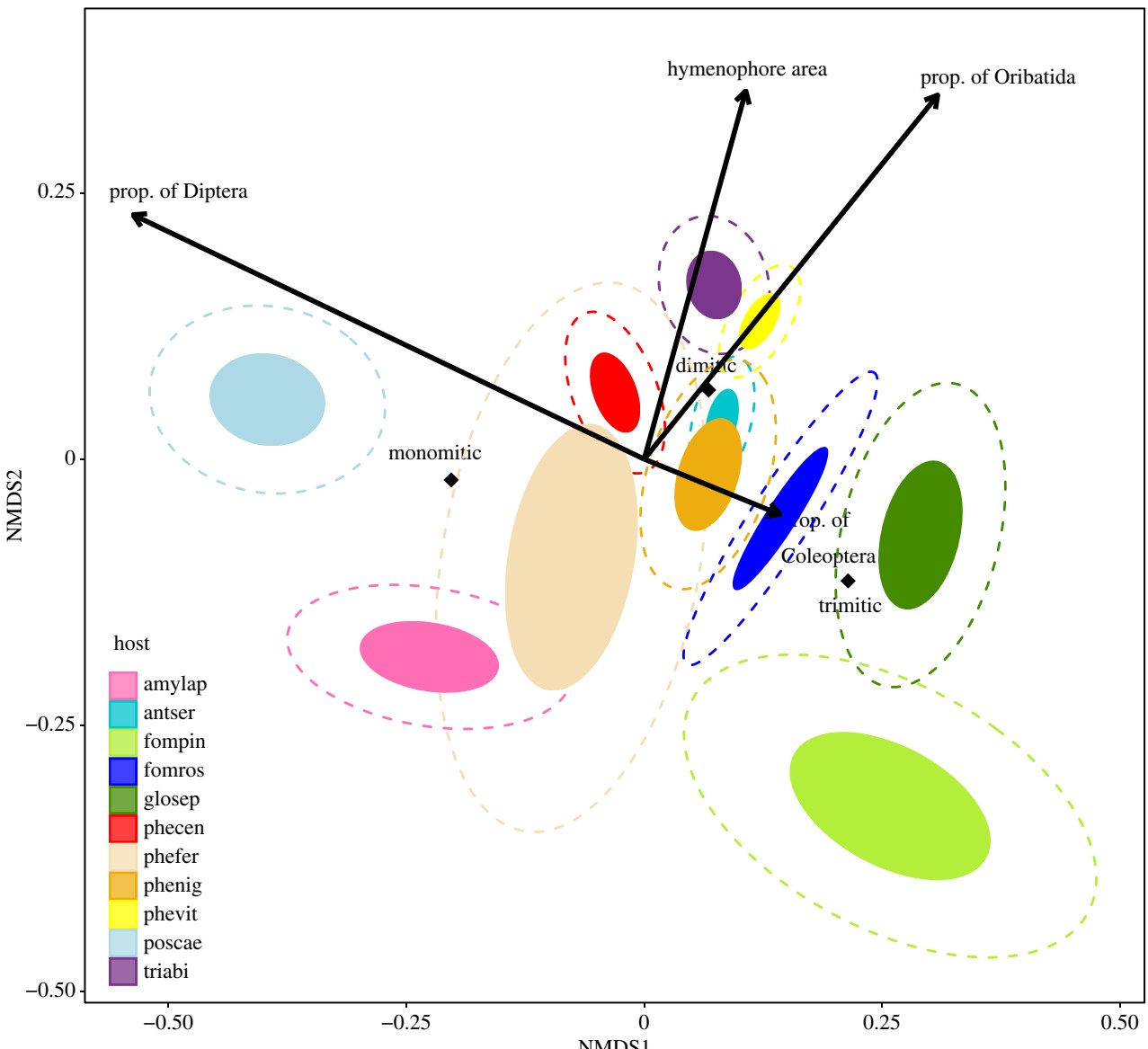

**Figure 2.** Ordination biplot of arthropod OTUs amplified from 11 species of fungal hosts. Fungal hosts are grouped by confidence interval levels with filled inner circles representing the 0.5 level and dotted outer circles the 0.75 level. The ordination is based on a global NMDS (stress = 0.114, $k = 4$) of Bray–Curtis dissimilarities from 180 fruit body samples. The axes are sorted by most variation explained and scaled in (half-change) units of compositional turnover. Vectors and centroids are fitted with the envfit function (in package VEGAN) and were significant in multivariate tests. Proportions of arthropod orders were calculated per sample from the dataset, not from species scores.

points for colonization, we should nonetheless consider the possibility of biased sampling between the two fruit body morphologies; resupinates are thinner implying a higher risk of contamination from arthropods which have merely walked or grazed on the surface of the fruit body.

Arthropod diversity was not affected by persistence or toughness, which are collinear variables. As long-lived fruit bodies are habitats with more temporal stability, we expected a higher diversity because species would have more time to accumulate. On the other hand, if the presence of arthropods reduces fungal fitness, the fungus could invest in chemical or physical defence to mitigate these effects. Guevara & Rayner [72] found that fruit body grazing by two fungivorous beetles had a negative effect on reproduction of the polypore *Trametes versicolor*. Tough fruit bodies are probably harder for an arthropod to gnaw in and act as a physical defence [21]. Whether these fungi also invest in chemical defence has not yet been asserted, although there are some indications [24,72].

Koskinen & Roslin [29] also used DNA metabarcoding to study arthropod communities in fruit bodies of short-lived agaric mushrooms. They found a different faunal composition which was almost deprived of mites and largely dominated by short-horned flies (Brachycera), in contrast with a dominance of thread-horned flies (nematocerans) and mites in our study. A lower arthropod richness was detected, which might be due to inherent differences between fungi or methodological approaches between studies. Although both studies targeted part of the COI gene, the primer pair BF3/BR2 [45] amplified a fragment that was twice as long (greater than 400 bp), albeit not overlapping with, the primer pair used by Koskinen & Roslin [29]. The latter pair, ZBJ-Art [73], was designed for studies on gut contents of bats, while BF3/BR2 was optimized for detecting freshwater arthropods. It is thus difficult to compare the efficiency of these primers in detecting arthropod DNA from fungal samples, as none were designed to amplify arthropods in forest ecosystems.

**Table 3.** Environmental variables fitted onto a NMDS configuration describing community composition in arthropod OTUs. Axis 1 (NMDS1) and 2 (NMDS2) give the coordinates of the heads of vector arrows or factor averages. $R^2$ gives the correlation coefficient for each vector and factor on the NMDS. $p$-values are assessed through permutation tests (perm. = 999, $\alpha = 0.05$, significant values in italics). Continuous variables are fitted as vectors with maximum correlation to the configuration, i.e. the direction of most rapid change in ordination space. Categorical variables are fitted as averages of ordination scores for each factor level.

| environmental variables | NMDS1 | NMDS2 | $R^2$ | $p$ |
|---|---|---|---|---|
| *vectors* | | | | |
| thickness (mean) | 0.0145 | −0.9999 | 0.3039 | *0.001* |
| hymenophore area | 0.2950 | 0.9555 | 0.1543 | *0.001* |
| *factors* | | | | |
| *hyphal system* | | | 0.3443 | *0.001* |
| monomitic | −0.1103 | −0.0212 | | |
| dimitic | 0.0735 | 0.0704 | | |
| trimitic | 0.2326 | −0.1241 | | |
| *morphology* | | | 0.0941 | *0.019* |
| pileate | −0.0681 | −0.0899 | | |
| resupinate | 0.0521 | 0.0688 | | |
| *size* | | | 0.3004 | *0.001* |
| size 1 | −0.3960 | −0.0559 | | |
| size 2 | 0.0851 | 0.0611 | | |
| size 3 | 0.0021 | −0.1299 | | |

## (b) Are arthropod communities shaped by fruit body traits?

As predicted, the arthropod community was structured according to fruit body persistence and toughness. While the high proportions of true flies were related to soft and short-lived fruit bodies, high proportions of beetles and oribatid mites were associated with tough and long-lived fruit bodies. Arthropods, which develop, feed or shelter in living fruit bodies of fungi, might distinguish between the habitat's persistence. Lacy [22] showed that persistence of the fungal fruit body affected the host choice of drosophilid flies. From rearing-based studies, arthropod communities in fruit bodies of tough wood-decay fungi clearly differ from those of the soft and short-lived agaric mushrooms [25,28,34]. True flies dominate the latter [27–29], perhaps because they can develop rapidly matching the short lifespan of the mushrooms [30]. On the other hand, beetles and oribatid mites, which typically need longer development times [30,31], were more common in long-lived fruit bodies. The fact that toughness of the fruit body affects community composition seems plausible and has earlier been found to structure beetle preferences to fungal hosts [18,21,37]. In leaf herbivores, species from separate feeding guilds may respond differently to toughness, due to various life histories or feeding strategies [9,74]. It is well-known from plant-herbivore systems that leaf toughness is one of the plant's major defences against insect herbivory [10,75,76]. Some, however, have convergently adapted to eat different types of tough leaves, for example, through modifications of their chewing mouthparts [77]. Fungivorous beetles also have mouthparts adapted for feeding on different macrofungal resources [78], although no studies have so far compared mouthparts of beetles from different wood-decay fungal hosts. We should nevertheless be careful about inferring a causal relationship between arthropod community differentiation and toughness. Hyphal systems can also be correlated with other factors, such as persistence or differing ability to hold water or nutrients.

Overall, the fungal fruit bodies hosted arthropod communities belonging to taxonomic groups we expected to be present in wood-decay fungi. The most common families we identified consist mostly (Bolitophilidae, Mycetophilidae and Oppidae) or partly (Chironomidae and Cecidomyiidae) of fungivorous species [27,79,80]. However, fungivores may not necessarily be dominant, as our annotations are largely unresolved at the species level and predators could make up to a third of the faunal composition in fruit bodies [25,39]. At the order level, true flies and oribatid mites were dominant, which we expected based on earlier rearing-based studies on living fruit bodies of wood-decay fungi [17,26,27,34]. It was, however, surprising that we did not find more springtails or beetles, as has been shown before [25,26,35].

## (c) Are there species-specific co-occurrences between arthropods and fungal hosts?

Nearly three-fourths of all arthropods were specific to one or two fungal hosts, a pattern that was particular in insects. In general, arthropods seemed to be specialized to one fungal host, to softer fruit bodies (i.e. mono- or dimitic hyphal systems), or not having any preference to toughness at all. Studies from mushrooms and their fungivores have found them to be largely unspecific in their host choice [22,29,81]. This could be due to mushrooms investing less in chemical and physical defence for their short-lived fruit bodies [26,34]. Here, there seems to be a relatively high degree of specialization among arthropods, which is supported by other studies from fruit bodies of wood-decay fungi [24,32,82]. Plants have evolved a wide array of physical and chemical defences to deter herbivores [83] which, in turn, selects for herbivore specialization towards plant hosts [84]. We found that some arthropods preferred softer fruit bodies, which indicates that fruit body toughness is related to arthropod host selection. Further, the high species-specificity we detected could be due to different chemical profiles between fungal hosts but, as we have not measured fruit body chemistry, these are mere speculations.

Several associations known from rearing-based studies were confirmed in the co-occurrence network. For example, the specialist tineid moth *Agnathosia mendiella* in *Fomitopsis rosea* [32], and mordellid beetle *Curtimorda* sp. in *Gloeophyllum sepiarium* [85]; several genera which have been found in *Fomitopsis pinicola*, like the predators *Lasioseius* sp. (a mesostigmatid mite) [86], *Lestodiplopsis* sp. (a gall midge) [87] and the—presumably fungivorous—oribatid mite *Carabodes* sp. [35,88], which also co-occurred with *G. sepiarium*, *Phlebia centrifuga* and *Trichaptum abietinum*.

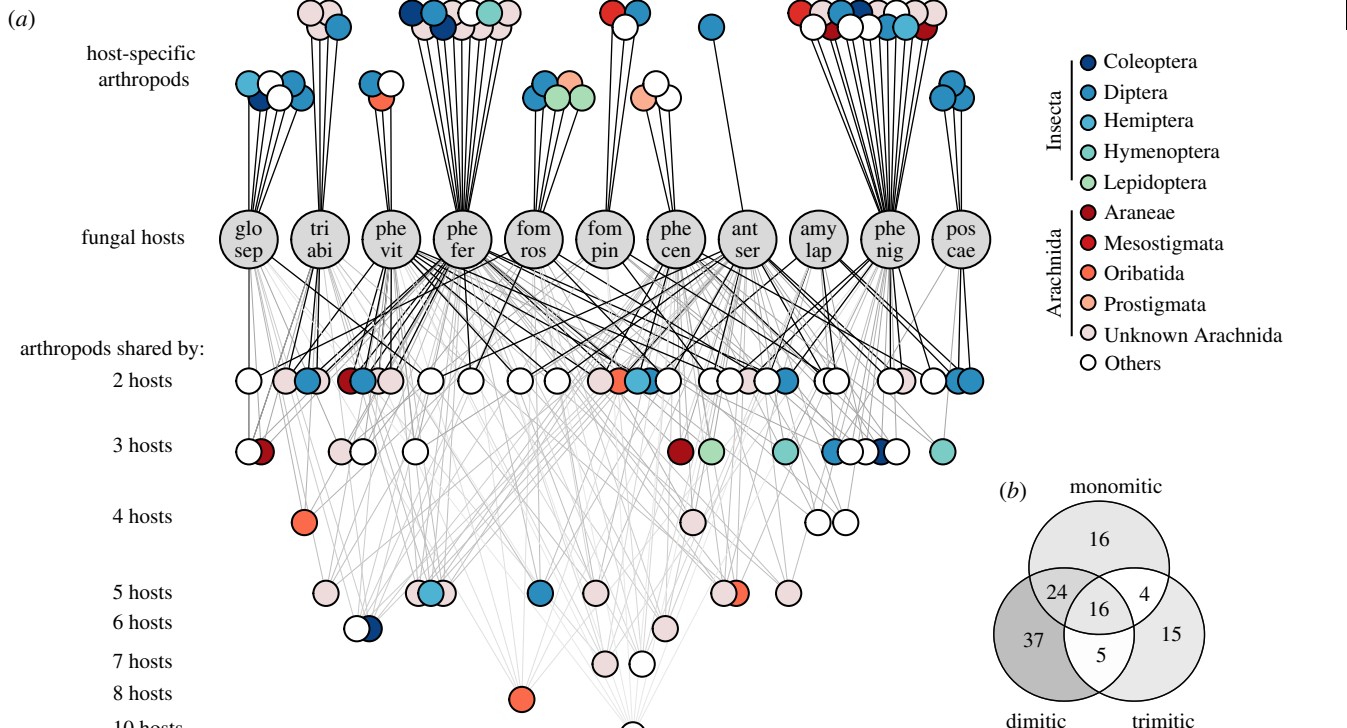

**Figure 3.** Co-occurrence relationships between 117 arthropod OTUs and 11 fungal hosts. (*a*) Tripartite network illustrating co-occurrence relationships between arthropod OTUs and fungal hosts (grey circles). Circles at the top represent arthropods co-occurring with only one fungal host. Circles underneath represent arthropods co-occurring with 2–10 fungal hosts. Arthropods are coloured by orders, those within class Arachnida in red and Insecta in blue or green. (*b*) Venn diagram showing the number of arthropod OTUs co-occurring with fungal hosts of differing hyphal systems: mono-, di- or trimitic. Co-occurrences are defined as significant associations from a multi-level pattern analysis based on indicator values. Network is visualized by IGRAPH and layout is manually adjusted from a tripartite Sugiyama layout algorithm.

## 5. Conclusion and perspectives

By applying DNA metabarcoding to explore arthropods in different fungi, we filled an important knowledge gap on how arthropod communities are structured in *living* fungal fruit bodies, which is an understudied system relative to dead fruit bodies. The species network clearly revealed several co-occurrences that are known from rearing-based studies, thus supporting taxon identification with a DNA approach.

Maurice & Arnault [40] also found a higher diversity of fungicolous fungi in species with resupinate fruit bodies. Given that both studies analysed the same samples, there are indications that fungicolous fungi and arthropods might respond to the same fruit body traits. The separation of host fungi in ordination space is similar between the two studies. Could the presence of fungicolous fungi be affected by arthropods, or *vice versa*? Beetles in dead wood habitats can vector diverse communities of fungal DNA, although antagonistic and indirect interactions could also be important.

As discussed above, plants invest in a wide array of defence mechanisms to deter or reduce arthropod grazing, for example, secondary metabolites or tough leaves. Compared to the well-documented antagonistic relationships between plants and herbivores, the paucity of studies on interactions between fungi and fungivores is striking. We found that most fruit bodies of wood-decay fungi hosted a distinct arthropod fauna, with overall a relatively high level of host specificity among them. Although the reasons behind this specificity may be manifold, fungi most likely invest in chemical and physical defences against fungivores. Guevara & Rayner [72] showed that fungivory could reduce

reproductive fitness in *Trametes versicolor*, while Jonsell & Nordlander [24] suggested that high host specificity could be explained by host chemistry, and our results indicate that physical defence, i.e. toughness of the fruit body, is a potential constraint to arthropods. We call for improving our understanding of fungivory effects on fungal fitness, in particular the physical and chemical defence against arthropod grazing.

**Data accessibility.** The MiSeq raw sequences are available on NCBI SRA SUB10678225. The OTU table, metadata and intermediate files are available from the Dryad Digital Repository: https://doi.org/10.5061/dryad.b5mkkwhdv [89], and R scripts for the figures and analyses are in Zenodo (https://doi.org/10.5281/zenodo.5503516).

**Authors' contributions.** L.L.: formal analysis, methodology, visualization and writing—original draft; T.B.: conceptualization, funding acquisition, resources, supervision, validation and writing—review and editing; H.K.: conceptualization, resources, supervision and writing—review and editing; L.B.: writing—review and editing; R.M.J.: writing—review and editing; L.M.: methodology, software, writing—review and editing; A.S.: writing—review and editing; S.M.: conceptualization, formal analysis, funding acquisition, methodology, project administration, resources, supervision, visualization and writing—review and editing.

All authors gave final approval for publication and agreed to be held accountable for the work performed therein.

**Competing interests.** We declare that we have no competing interests.

**Funding.** This study was supported in part by the Research Council of Norway (grant no. 254746 to S.M. and H.K.).

**Acknowledgements.** We acknowledge Vasco Elbrecht and Jordan Cuff for discussions on primer choice to amplify arthropods, Douglas Yu for sharing barcode sequences, and Pedro Maria Martin-Sanchez and Eva Lena Estensmo for providing gDNA of *C. longicaudata*.

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
