## [Peer Review File · Proceedings of the Royal Society B: Biological Sciences]

Review History

RSPB-2021-2020.R0 (Original submission)

Review form: Reviewer 1

Recommendation

Major revision is needed (please make suggestions in comments)

Scientific importance: Is the manuscript an original and important contribution to its field?

Good

General interest: Is the paper of sufficient general interest?

Good

Quality of the paper: Is the overall quality of the paper suitable?

Good

Is the length of the paper justified?

Yes

Should the paper be seen by a specialist statistical reviewer?

No

Do you have any concerns about statistical analyses in this paper? If so, please specify them explicitly in your report.

Yes

It is a condition of publication that authors make their supporting data, code and materials available - either as supplementary material or hosted in an external repository. Please rate, if applicable, the supporting data on the following criteria.

Is it accessible?

No

Is it clear?

N/A

Is it adequate?

N/A

Do you have any ethical concerns with this paper?

No

Comments to the Author

The manuscript, "DNA metabarcoding reveals host-specific communities of arthropods residing in fungal fruit bodies" by Lunde et al. provide insight into the diversity and structure of arthropod communities hosted by fungal fruit bodies. The findings presented in this paper are focused on determining if arthropod alpha diversity metrics are impacted by fungal fruit traits, how arthropod communities are structured depending on fungal traits, and if arthropod communities display host specificity which may be influenced by host traits. Fungal fruit traits (6) were collected for 11 species of wood decay fungus (19-26 samples each) along with excised internal fungal tissue for amplicon sequencing analysis. The authors report that fungal morphology had a significant effect on alpha diversity, with highest diversity found in resupinate fungi. They also report that certain fungal traits (i.e., toughness, persistence) had significant effects on the arthropod community structure, with true flies more commonly associated with soft short-lived fungi, while beetles and oribatid mites more closely associated with tougher fruiting fungi. Finally, authors report that a majority of arthropod communities showing host specificity. Overall, the paper presents interesting findings that add to our ecological knowledge of fungal host traits and their effect on the diversity and structure of associated arthropod communities.

Some significant concerns include: the lack of information about the global NMDS method, the model selection approach in evaluating the effect of host traits on alpha diversity, and potential contamination issues in resupinate species.

The authors discuss using a global NMDS method, but do not give information about why this method was chosen, and also it is confusing what they mean by global NMDS. The vegan package documentation (<https://search.r-project.org/CRAN/refmans/vegan/html/monoMDS.html>) says the default, normal NMDS is "global" but here the authors are clearly doing something differently, and their ordination has stress levels that are much higher than one would normally see for NMDS.

I wanted to review the code to see how the authors did the global NMDS analysis, but I only found the link for the Dryad repository when I went to submit my review. When I tried to download the code, my only option was to download the full 5 GB zipped folder. I'm not willing to do this just to access the R code file, and I think it should be made much more easily accessible to reviewers. The sequencing data should be deposited to NCBI SRA or a similar sequence repository, and only the OTU table and statistical analysis (in this case the R code) and any other

necessary metadata and data tables should be included in the Dryad (or similar) repository. A link to the data and analysis code should be included in the manuscript.

Other comments:

Line 27: Word choice: “characterized” might be a better option than “revealed”

Lines 48-50: Please include a citation supporting the statement about how many arthropods are sustained on single food items and how these represent discrete habitat patches.

Lines 54-55: If comparing beetles to beetles, be specific about the type of beetle in line 54.

Line 57: The study does a great job highlighting effects of host traits on arthropod diversity and community composition. It’s not clear that this is a “trophic network” since there are only two levels and arthropods may use fungi as shelter/habitat and not necessarily as food. Perhaps “interaction” would be better.

Lines 85-90: Great way to show how your study fills a gap and addresses a larger question!

Line 95: For question 1, you should specify that you mean alpha diversity, and then for question 2 you should specify beta diversity. This makes it clearer what the differences are between your questions.

Line 119: How is fruit body size measured? From the supplement I can see that there are 3 levels: small, medium, and large. In future studies it would be good to have a more quantitative measurement for fruit body size.

Lines 122-123: Please include a citation for hyphal system complexity correlated to toughness.

Lines 129-130 & 279-282: How where you able to collect interior tissue that was not contaminated from resupinate species? As you mention, for very thin species, being able to aseptically excise interior tissue to avoid contamination from transient species or grazers is critical to make conclusions about communities of arthropods hosted by these fungi. The weak but significant effect for high diversity in resupinate species is harder to accept due to this.

Lines 183-188: What is the justification for using 5 univariate models and information criteria instead of a multivariate model? Did you test for collinearity? A better approach might be to use a multivariate model for uncorrelated variables to acknowledge multiple factors that might be influencing the response. If unable to use a multivariate model because of collinearity, a measure of model fit should be included to show support for your chosen model.

Line 190: alpha set to 0.5 (shouldn’t this be 0.05?)

Lines 268-270: Help support this section by reminding readers which of the fungal morphology types are homogeneous and heterogeneous.

Line 314: Slightly confusing wording.

Line 322: Slightly confusing wording. Consider, “as hyphal systems can ALSO be correlated with other important factors, such as persistence or DIFFERING ability to hold water or nutrients”

Tables: The commas in the numbers is confusing for me (although I acknowledge that it is more common in parts of Europe)

Figures:

Figure 1: To make this paper more accessible to a general audience, I suggest adding representative photographs/images of resupinate vs pileate species here. Clustering your species by type and not alphabetically (for panels 1a/c) would help a reader see patterns in the data.

Figure 2 legend, Line 614: How do these stress values relate to ones in “normal” NMDS?

Figure 2: Showing your actual data points in the NMDS would be more informative.

For either figure 2 or 3 it would be good to include an image depicting monomictic, dimictic and trimictic morphology

Figure 4: It would be nice to see some specific examples highlighted here.

Supplementary Figure 5: Your chosen rarefaction cutoff for this paper is fine, but I want to clarify that you do not want to choose a cutoff below the asymptote saturation. It is much better if you have enough sequencing depth to choose a cutoff after your alpha diversity has saturated. The text is a bit confusing here.

Review form: Reviewer 2

Recommendation

Accept with minor revision (please list in comments)

Scientific importance: Is the manuscript an original and important contribution to its field?

Excellent

General interest: Is the paper of sufficient general interest?

Good

Quality of the paper: Is the overall quality of the paper suitable?

Excellent

Is the length of the paper justified?

Yes

Should the paper be seen by a specialist statistical reviewer?

No

Do you have any concerns about statistical analyses in this paper? If so, please specify them explicitly in your report.

No

It is a condition of publication that authors make their supporting data, code and materials available - either as supplementary material or hosted in an external repository. Please rate, if applicable, the supporting data on the following criteria.

Is it accessible?

No

Is it clear?

No

Is it adequate?

Yes

Do you have any ethical concerns with this paper?

No

Comments to the Author

This is an interesting and novel study examining the communities of arthropods associated with the fruiting bodies of wood decay fungi. With the rise in the number of studies utilizing metabarcoding to investigate hidden diversity, it is surprising that the diversity within living fungi has been neglected in contrast to the extensive literature on plant-associated communities. This study begins to fill in that gap and points to exciting directions for future work. How analogous are fungus-arthropod interactions to the multifaceted ecological and evolutionary processes known from plant-arthropod interactions? This paper is an important first step. Not only do the authors reveal the hidden diversity of fungus-associated arthropods, but they also address some interesting more general questions, relating to the relationship between host specialization and ephemerality of the resource, how host traits influence their communities, and how habitat size influences community assembly. The counter-intuitive association between fruiting body size and arthropod diversity is noteworthy; unlike with larger ecosystems, host-associated symbiotic communities, to my knowledge, do not consistently follow the rules of island biogeography theory, so this is another example where the expectation from theory developed from larger organisms is not met. The level of host specificity in the OTUs observed is fascinating and opens up many questions as to the factors selecting for and maintaining these interactions. Finally, not only is this study an important contribution based on the particular subject matter, but I believe the methods are of note for metabarcoding approaches. Namely, a convincing case was made for the value of actually testing whether ASVs or OTUs are the more appropriate approach for the particular system in question. In short, the subject matter is original, the methods are sound and rigorous, and the results and conclusions are noteworthy. I mostly only have relatively minor suggestions:

Page 4 line 73: Do "predators" and "parasites" here refer to arthropods which predate on other arthropods and parasitoids, respectively? The use of "parasite" is particularly ambiguous as it could conceivably relate to the nature of the interaction between host fungus and symbiotic arthropod (as opposed to mutualistic). Using "parasitoid" at least would make clear the point that this is an ecosystem with multiple trophic levels beyond the direct fungus-arthropod interaction. Also, there's no discussion of the potential ecological roles of the observed OTUs later on. Are the patterns of diversity you observe driven by fungivores or are OTUs belonging to (the probably more transiently-associated) category of predators also important for driving certain trends. [see comment about the caveat on page 9 below, this is a related point]

Page 7 line 170: A strong methodological argument for using OTUs over ASVs in this case.

Page 7 line 173: It would be of interest to report what percentage of OTUs were annotated to fungi. Was it close to the remaining 54%?

Page 9 line 222: I find it quite surprising that a fifth of the sequences could not be identified below the level of Arthropoda. Is this a consequence of a limited reference database (does it not contain all known orders of arthropods?). Alternatively, do you suspect there's a major risk of degradation prior to extraction – might enzymes inside the fungal fruiting bodies decrease the integrity of the DNA within the sample?

Page 9 lines 279-282: Could this caveat be addressed by comparing the results of metabarcoding the outer vs. inner layers of pileate fruit bodies? Presumably, the risk of 'contamination' from transient visitors should be more apparent on the outer surfaces, especially if there are taxa that are seen on resupinate fungi and the outer surfaces of pileate ones, but not in the interiors of the pileate fungi. Also, if such contamination is at play it should be apparent from examining the particular taxonomic composition of your resupinate samples. Based on the most frequent families you report, it appears that the major taxa you've observed are mostly arthropods which

are known to dwell in fungi. It would be quite telling if the OTUs driving the higher diversity in resupinate samples are annotated as less clearly fungus-associated taxa. Can you do an ANCOM test to determine which are the OTUs that occur only in your resupinate samples?

Page 10 line 290: This will be an especially interesting topic of future investigation, along with just generally investigating the role of physiological characteristics of the living fungal hosts.

Figure 1: Is there any way to indicate which species are pileate and which are resupinate on this figure? Perhaps you can design small representative icons to place right next to the abbreviated species names. You'd also be able to easily fit the key directly above or below the taxonomic key for the stacked barplot. Or maybe even more simply, just change the color coding for the diversity barplots in (1a) to match the color code for (1b). As it stands, assigning each individual species a separate color does not add much information (other than creating continuity with Fig. 2, but the key in that figure suffices). It would carry more information to have open boxes for pileate species and solid grey for resupinate in (1a).

Decision letter (RSPB-2021-2020.R0)

09-Nov-2021

Dear Ms Lunde,

I am writing to inform you that your manuscript RSPB-2021-2020 entitled "DNA metabarcoding reveals host-specific communities of arthropods residing in fungal fruit bodies" has, in its current form, been rejected for publication in Proceedings B.

This action has been taken on the advice of referees, who have recommended that substantial revisions are necessary. With this in mind we would be happy to consider a resubmission, provided the comments of the referees are fully addressed. However please note that this is not a provisional acceptance.

- 1) A 'response to referees' document including details of how you have responded to the comments, and the adjustments you have made.
- 2) A clean copy of the manuscript and one with 'tracked changes' indicating your 'response to referees' comments document.
- 3) Line numbers in your main document.
- 4) Data - please see our policies on data sharing to ensure that you are complying (<https://royalsociety.org/journals/authors/author-guidelines/#data>). Please also take note of the reviewer's comments below with regard to providing readily accessible data files.

To upload a resubmitted manuscript, log into <http://mc.manuscriptcentral.com/prsb> and enter your Author Centre, where you will find your manuscript title listed under "Manuscripts with

Decisions." Under "Actions," click on "Create a Resubmission." Please be sure to indicate in your cover letter that it is a resubmission, and supply the previous reference number.

Sincerely,
 Professor Loeske Kruuk
 mailto: proceedingsb@royalsociety.org

Associate Editor
 Board Member: 1
 Comments to Author:

Two expert referees find considerable value in this study, and that it has clear potential to make an interesting and important contribution to the field, as well as of wider biological interest. However, both reviewers flag up some issues of methodology. These are mostly along the lines of providing clearer explanation and justification of the approach, and this is definitely important to address. These, as well as all other reviewer comments, should be carefully considered in any revised manuscript.

Reviewer(s)' Comments to Author:

Referee: 1

Comments to the Author(s)

The manuscript, "DNA metabarcoding reveals host-specific communities of arthropods residing in fungal fruit bodies" by Lunde et al. provide insight into the diversity and structure of arthropod communities hosted by fungal fruit bodies. The findings presented in this paper are focused on determining if arthropod alpha diversity metrics are impacted by fungal fruit traits, how arthropod communities are structured depending on fungal traits, and if arthropod communities display host specificity which may be influenced by host traits. Fungal fruit traits (6) were collected for 11 species of wood decay fungus (19-26 samples each) along with excised internal fungal tissue for amplicon sequencing analysis. The authors report that fungal morphology had a significant effect on alpha diversity, with highest diversity found in resupinate fungi. They also report that certain fungal traits (i.e., toughness, persistence) had significant effects on the arthropod community structure, with true flies more commonly associated with soft short-lived fungi, while beetles and oribatid mites more closely associated with tougher fruiting fungi. Finally, authors report that a majority of arthropod communities showing host specificity. Overall, the paper presents interesting findings that add to our ecological knowledge of fungal host traits and their effect on the diversity and structure of associated arthropod communities.

Some significant concerns include: the lack of information about the global NMDS method, the model selection approach in evaluating the effect of host traits on alpha diversity, and potential contamination issues in resupinate species.

The authors discuss using a global NMDS method, but do not give information about why this method was chosen, and also it is confusing what they mean by global NMDS. The vegan package documentation (<https://search.r-project.org/CRAN/refmans/vegan/html/monoMDS.html>) says the default, normal NMDS is "global" but here the authors are clearly doing something differently, and their ordination has stress levels that are much higher than one would normally see for NMDS.

I wanted to review the code to see how the authors did the global NMDS analysis, but I only found the link for the Dryad repository when I went to submit my review. When I tried to download the code, my only option was to download the full 5 GB zipped folder. I'm not willing to do this just to access the R code file, and I think it should be made much more easily accessible to reviewers. The sequencing data should be deposited to NCBI SRA or a similar sequence repository, and only the OTU table and statistical analysis (in this case the R code) and any other

necessary metadata and data tables should be included in the Dryad (or similar) repository. A link to the data and analysis code should be included in the manuscript.

Other comments:

Line 27: Word choice: “characterized” might be a better option than “revealed”

Lines 48-50: Please include a citation supporting the statement about how many arthropods are sustained on single food items and how these represent discrete habitat patches.

Lines 54-55: If comparing beetles to beetles, be specific about the type of beetle in line 54.

Line 57: The study does a great job highlighting effects of host traits on arthropod diversity and community composition. It’s not clear that this is a “trophic network” since there are only two levels and arthropods may use fungi as shelter/habitat and not necessarily as food. Perhaps “interaction” would be better.

Lines 85-90: Great way to show how your study fills a gap and addresses a larger question!

Line 95: For question 1, you should specify that you mean alpha diversity, and then for question 2 you should specify beta diversity. This makes it clearer what the differences are between your questions.

Line 119: How is fruit body size measured? From the supplement I can see that there are 3 levels: small, medium, and large. In future studies it would be good to have a more quantitative measurement for fruit body size.

Lines 122-123: Please include a citation for hyphal system complexity correlated to toughness.

Lines 129-130 & 279-282: How were you able to collect interior tissue that was not contaminated from resupinate species? As you mention, for very thin species, being able to aseptically excise interior tissue to avoid contamination from transient species or grazers is critical to make conclusions about communities of arthropods hosted by these fungi. The weak but significant effect for high diversity in resupinate species is harder to accept due to this.

Lines 183-188: What is the justification for using 5 univariate models and information criteria instead of a multivariate model? Did you test for collinearity? A better approach might be to use a multivariate model for uncorrelated variables to acknowledge multiple factors that might be influencing the response. If unable to use a multivariate model because of collinearity, a measure of model fit should be included to show support for your chosen model.

Line 190: alpha set to 0.5 (shouldn’t this be 0.05?)

Lines 268-270: Help support this section by reminding readers which of the fungal morphology types are homogeneous and heterogeneous.

Line 314: Slightly confusing wording.

Line 322: Slightly confusing wording. Consider, “as hyphal systems can ALSO be correlated with other important factors, such as persistence or DIFFERING ability to hold water or nutrients”

Tables: The commas in the numbers is confusing for me (although I acknowledge that it is more common in parts of Europe)

Figures:

Figure 1: To make this paper more accessible to a general audience, I suggest adding representative photographs/images of resupinate vs pileate species here. Clustering your species by type and not alphabetically (for panels 1a/c) would help a reader see patterns in the data.

Figure 2 legend, Line 614: How do these stress values relate to ones in “normal” NMDS?
 Figure 2: Showing your actual data points in the NMDS would be more informative.

For either figure 2 or 3 it would be good to include an image depicting monomictic, dimictic and trimictic morphology

Figure 4: It would be nice to see some specific examples highlighted here.

Supplementary Figure 5: Your chosen rarefaction cutoff for this paper is fine, but I want to clarify that you do not want to choose a cutoff below the asymptote saturation. It is much better if you have enough sequencing depth to choose a cutoff after your alpha diversity has saturated. The text is a bit confusing here.

Referee: 2

Comments to the Author(s)

This is an interesting and novel study examining the communities of arthropods associated with the fruiting bodies of wood decay fungi. With the rise in the number of studies utilizing metabarcoding to investigate hidden diversity, it is surprising that the diversity within living fungi has been neglected in contrast to the extensive literature on plant-associated communities. This study begins to fill in that gap and points to exciting directions for future work. How analogous are fungus-arthropod interactions to the multifaceted ecological and evolutionary processes known from plant-arthropod interactions? This paper is an important first step. Not only do the authors reveal the hidden diversity of fungus-associated arthropods, but they also address some interesting more general questions, relating to the relationship between host specialization and ephemerality of the resource, how host traits influence their communities, and how habitat size influences community assembly. The counter-intuitive association between fruiting body size and arthropod diversity is noteworthy; unlike with larger ecosystems, host-associated symbiotic communities, to my knowledge, do not consistently follow the rules of island biogeography theory, so this is another example where the expectation from theory developed from larger organisms is not met. The level of host specificity in the OTUs observed is fascinating and opens up many questions as to the factors selecting for and maintaining these interactions. Finally, not only is this study an important contribution based on the particular subject matter, but I believe the methods are of note for metabarcoding approaches. Namely, a convincing case was made for the value of actually testing whether ASVs or OTUs are the more appropriate approach for the particular system in question. In short, the subject matter is original, the methods are sound and rigorous, and the results and conclusions are noteworthy. I mostly only have relatively minor suggestions:

Page 4 line 73: Do "predators" and "parasites" here refer to arthropods which predate on other arthropods and parasitoids, respectively? The use of “parasite” is particularly ambiguous as it could conceivably relate to the nature of the interaction between host fungus and symbiotic arthropod (as opposed to mutualistic). Using "parasitoid" at least would make clear the point that this is an ecosystem with multiple trophic levels beyond the direct fungus-arthropod interaction. Also, there's no discussion of the potential ecological roles of the observed OTUs later on. Are the patterns of diversity you observe driven by fungivores or are OTUs belonging to (the probably more transiently-associated) category of predators also important for driving certain trends. [see comment about the caveat on page 9 below, this is a related point]

Page 7 line 170: A strong methodological argument for using OTUs over ASVs in this case.

Page 7 line 173: It would be of interest to report what percentage of OTUs were annotated to fungi. Was it close to the remaining 54%?

Page 9 line 222: I find it quite surprising that a fifth of the sequences could not be identified below the level of Arthropoda. Is this a consequence of a limited reference database (does it not contain all known orders of arthropods?). Alternatively, do you suspect there's a major risk of degradation prior to extraction – might enzymes inside the fungal fruiting bodies decrease the integrity of the DNA within the sample?

Page 9 lines 279-282: Could this caveat be addressed by comparing the results of metabarcoding the outer vs. inner layers of pileate fruit bodies? Presumably, the risk of 'contamination' from transient visitors should be more apparent on the outer surfaces, especially if there are taxa that are seen on resupinate fungi and the outer surfaces of pileate ones, but not in the interiors of the pileate fungi. Also, if such contamination is at play it should be apparent from examining the particular taxonomic composition of your resupinate samples. Based on the most frequent families you report, it appears that the major taxa you've observed are mostly arthropods which are known to dwell in fungi. It would be quite telling if the OTUs driving the higher diversity in resupinate samples are annotated as less clearly fungus-associated taxa. Can you do an ANCOM test to determine which are the OTUs that occur only in your resupinate samples?

Page 10 line 290: This will be an especially interesting topic of future investigation, along with just generally investigating the role of physiological characteristics of the living fungal hosts.

Figure 1: Is there any way to indicate which species are pileate and which are resupinate on this figure? Perhaps you can design small representative icons to place right next to the abbreviated species names. You'd also be able to easily fit the key directly above or below the taxonomic key for the stacked barplot. Or maybe even more simply, just change the color coding for the diversity barplots in (1a) to match the color code for (1b). As it stands, assigning each individual species a separate color does not add much information (other than creating continuity with Fig. 2, but the key in that figure suffices). It would carry more information to have open boxes for pileate species and solid grey for resupinate in (1a).

Author's Response to Decision Letter for (RSPB-2021-2020.R0)

See Appendix A.

RSPB-2021-2622.R0

Review form: Reviewer 1

Recommendation

Accept with minor revision (please list in comments)

Scientific importance: Is the manuscript an original and important contribution to its field?

Good

General interest: Is the paper of sufficient general interest?

Good

Quality of the paper: Is the overall quality of the paper suitable?

Good

Is the length of the paper justified?

Yes

Should the paper be seen by a specialist statistical reviewer?

Yes

Do you have any concerns about statistical analyses in this paper? If so, please specify them explicitly in your report.

Yes

It is a condition of publication that authors make their supporting data, code and materials available - either as supplementary material or hosted in an external repository. Please rate, if applicable, the supporting data on the following criteria.

Is it accessible?

Yes

Is it clear?

Yes

Is it adequate?

Yes

Do you have any ethical concerns with this paper?

No

Comments to the Author

As noted in my previous review, I believe this paper presents interesting findings that add to our ecological knowledge of fungal host traits and their effect on the diversity and structure of associated arthropod communities. I value the combination of traditional mycological approaches using morphology coupled with diversity metrics using sequencing data.

The authors have addressed my main concerns by: correcting the stress values for the NMDS and adding more information about the approach, clarifying why they chose to use univariate models, adding photographs in the supplementary material, and making the data more accessible to reviewers and readers. They also re-organized the species in Figure 1 to show the results more clearly.

The authors should also specify that the R code is deposited in Zenodo and include the link in the text under "Data accessibility" or just include the R code in the Dryad folder.

I'm confused as to why the authors chose to use a global version of monoMDS (and then doing a Procrustes analysis) instead of using metaMDS, which does all this for you while also using several random starts. Either way, I find the "gNMDS" designation misleading because, as the authors confirm in their response, they are using what is considered the normal form of NMDS. I think it is OK to state that it is a global NMDS when first introduced, but all the "gNMDS" should be changed to simple "NMDS" to align with norms and other publications. If the analysis is indeed different in some way than it would be with metaMDS, then the authors should justify their approach and state how it is different.

Overall, I appreciate the correction of typos and terminology.

Decision letter (RSPB-2021-2622.R0)

23-Dec-2021

Dear Ms Lunde

Some good news before the Christmas break...

I am pleased to inform you that your manuscript RSPB-2021-2622 entitled "DNA metabarcoding reveals host-specific communities of arthropods residing in fungal fruit bodies" has been accepted for publication in Proceedings B.

The referee and Associate Editor have recommended publication, but have also suggested some minor revisions to your manuscript. Therefore, I invite you to respond to the referee(s)' comments and revise your manuscript. The schedule for publication is very tight, and we usually ask that you submit the revised version of your manuscript within 7 days. However given the upcoming Christmas break, we realise this may not be feasible, so please get in touch if you would like an extension.

Once again, thank you for submitting your manuscript to Proceedings B. As this is one of my last editorial decision letters of the year, I'm very pleased to be sending positive news. We look forward to receiving your revision, but if you have any questions meanwhile, please get in touch.

Finally, all the best to you and your co-authors for Christmas and the New Year.

Yours sincerely,
Professor Loeske Kruuk
mailto: proceedingsb@royalsociety.org

Associate Editor
Board Member
Comments to Author:

I agree with the reviewer that the suggested revisions have been made to a high standard. A very small number of minor issues remain, which should be addressed.

Reviewer(s)' Comments to Author:
Referee: 1

Comments to the Author(s).

As noted in my previous review, I believe this paper presents interesting findings that add to our ecological knowledge of fungal host traits and their effect on the diversity and structure of associated arthropod communities. I value the combination of traditional mycological approaches using morphology coupled with diversity metrics using sequencing data.

The authors have addressed my main concerns by: correcting the stress values for the NMDS and adding more information about the approach, clarifying why they chose to use univariate models, adding photographs in the supplementary material, and making the data more accessible to reviewers and readers. They also re-organized the species in Figure 1 to show the results more clearly.

The authors should also specify that the R code is deposited in Zenodo and include the link in the text under "Data accessibility" or just include the R code in the Dryad folder.

I'm confused as to why the authors chose to use a global version of monoMDS (and then doing a Procrustes analysis) instead of using metaMDS, which does all this for you while also using several random starts. Either way, I find the "gNMDS" designation misleading because, as the authors confirm in their response, they are using what is considered the normal form of NMDS. I think it is OK to state that it is a global NMDS when first introduced, but all the "gNMDS" should be changed to simple "NMDS" to align with norms and other publications. If the analysis is indeed different in some way than it would be with metaMDS, then the authors should justify their approach and state how it is different.

Overall, I appreciate the correction of typos and terminology.

Author's Response to Decision Letter for (RSPB-2021-2622.R0)

See Appendix B.

Decision letter (RSPB-2021-2622.R1)

04-Jan-2022

Dear Ms Lunde

I am pleased to inform you that your manuscript entitled "DNA metabarcoding reveals host-specific communities of arthropods residing in fungal fruit bodies" has been accepted for publication in Proceedings B.

Data Accessibility section

Open Access

Paper charges

Sincerely,

Proceedings B

Appendix A

Comments to Author:

Two expert referees find considerable value in this study, and that it has clear potential to make an interesting and important contribution to the field, as well as of wider biological interest. However, both reviewers flag up some issues of methodology. These are mostly along the lines of providing clearer explanation and justification of the approach, and this is definitely important to address. These, as well as all other reviewer comments, should be carefully considered in any revised manuscript.

>>> Dear Editor,

thank you for considering our manuscript. We have revised it according to the referees' comments. We clarified the analytical concerns that they addressed and implemented detailed information and discussion directly into the manuscript and Supplementary Material that include: (1) more information about the ordination method (gNMDS) with a short discussion on the stress values, (2) a table and test showing multicollinearity of the explanatory variables and justification for univariate models, (3) pictures of resupinate and pileate morphologies and hyphal systems, (4) precisions on the chosen rarefaction cutoff, (5) new data has been published to NCBI SRA SUB10678225 and DRYAD (<https://doi.org/10.5061/dryad.b5mkkwhdv>).

The line numbers in the response letter refers to the new version of the manuscript.

Referee: 1

Comments to the Author(s)

The manuscript, "DNA metabarcoding reveals host-specific communities of arthropods residing in fungal fruit bodies" by Lunde et al. provide insight into the diversity and structure of arthropod communities hosted by fungal fruit bodies. The findings presented in this paper are focused on determining if arthropod alpha diversity metrics are impacted by fungal fruit traits, how arthropod communities are structured depending on fungal traits, and if arthropod communities display host specificity which may be influenced by host traits. Fungal fruit traits (6) were collected for 11 species of wood decay fungus (19-26 samples each) along with excised internal fungal tissue for amplicon sequencing analysis. The authors report that fungal morphology had a significant effect on alpha diversity, with highest diversity found in resupinate fungi. They also report that certain fungal traits (i.e., toughness, persistence) had significant effects on the arthropod community structure, with true flies more commonly associated with soft short-lived fungi, while beetles and oribatid mites more closely associated with tougher fruiting fungi. Finally, authors report that a majority of arthropod communities showing host specificity. Overall, the paper presents interesting findings that add to our ecological knowledge of fungal host traits and their effect on the diversity and structure of associated arthropod communities.

Some significant concerns include: the lack of information about the global NMDS method, the model selection approach in evaluating the effect of host traits on alpha diversity, and potential contamination issues in resupinate species.

>>> Thank you for providing constructive comments on our manuscript. We have tried to address most of them, which has overall improved the precision of our manuscript. We have taken the following measures to deal with the main concerns that you identified:

- brought additional information about the NMDS approach in Supplementary Material 6 and corrected the stress value,
- precised the measure of model fit for all univariate models, and provided information about multicollinearity in Supplementary Material 8, including correlation tests of all variables, and
- plotted Figure 1 according to resupinate/pileate morphology

The authors discuss using a global NMDS method, but do not give information about why this method was chosen, and also it is confusing what they mean by global NMDS. The vegan package documentation (<https://search.r-project.org/CRAN/refmans/vegan/html/monoMDS.html>) says the default, normal NMDS is “global” but here the authors are clearly doing something differently, and their ordination has stress levels that are much higher than one would normally see for NMDS.

>>> Thank you for pointing out the stress levels which would indeed have been high. We have erroneously reported the wrong values – the true stress value was 0.114. We have made correction in the text. Indeed, the global NMDS is the most ‘normal’ NMDS variant today and it is the one we have used in the gradient analysis. We added more precision on the gNMDS settings in Supplementary Material 6 and deposited the code in Dryad (“arthropoda_analyses.R”)

I wanted to review the code to see how the authors did the global NMDS analysis, but I only found the link for the Dryad repository when I went to submit my review. When I tried to download the code, my only option was to download the full 5 GB zipped folder. I’m not willing to do this just to access the R code file, and I think it should be made much more easily accessible to reviewers. The sequencing data should be deposited to NCBI SRA or a similar sequence repository, and only the OTU table and statistical analysis (in this case the R code) and any other necessary metadata and data tables should be included in the Dryad (or similar) repository. A link to the data and analysis code should be included in the manuscript.

>>> We fully agree with this comment about making data readily accessible to reviewers in early stage. For the revision, we have uploaded the sequence data to NCBI SRA SUB10678225 under Bioproject PRJNA68025: <https://www.ncbi.nlm.nih.gov/bioproject/PRJNA680258>. The scripts for performing the statistical analyses, together with the OTU tables and metadata are available on DRYAD (<https://doi.org/10.5061/dryad.b5mkkwhdv>)

Other comments:

Line 27: Word choice: “characterized” might be a better option than “revealed”

>>> Corrected.

“Using DNA metabarcoding, we characterised the arthropod communities in...” (line 27)

Lines 48-50: Please include a citation supporting the statement about how many arthropods are sustained on single food items and how these represent discrete habitat patches.

>>> Corrected.

“...that comprise a single food item [e.g. 6, 7].” (line 49)

Lines 54-55: If comparing beetles to beetles, be specific about the type of beetle in line 54.

>>> We added more precision to the sentences.

“Short-lived hosts select for strong dispersers with fast developmental times, for example the beetles *Diaperis boleti* and *Tetratoma fungorum* breeding in fungal fruit bodies [11, 12]. While, in the long-lived habitat of hollow trees, hermit beetles (*Osmoderma eremita*) develop slowly and have limited dispersal abilities [13, 14].” (line 53-56)

Line 57: The study does a great job highlighting effects of host traits on arthropod diversity and community composition. It’s not clear that this is a “trophic network” since there are only two levels and arthropods may use fungi as shelter/habitat and not necessarily as food. Perhaps “interaction” would be better.

>>> Indeed, that is a good point and has been corrected.

“...in a poorly explored interaction network...” (line 57-58)

Lines 85-90: Great way to show how your study fills a gap and addresses a larger question!

>>> Thank you.

Line 95: For question 1, you should specify that you mean alpha diversity, and then for question 2 you should specify beta diversity. This makes it clearer what the differences are between your questions.

>>> We have now specified the differences between alpha and beta diversity analyses in the ‘Statistical analyses’ section in the M&M. We are really keen in keeping the research questions straightforward, especially for the general audience targeted by Proceedings B.

“Testing of alpha diversity (Q1)...” (line 185-186)

“The relationships between arthropod composition and all variables (i.e. beta diversity; Q2)...” (line 196-197)

Line 119: How is fruit body size measured? From the supplement I can see that there are 3 levels: small, medium, and large. In future studies it would be good to have a more quantitative measurement for fruit body size.

>>> In short, we characterised fruit body size in three classes, with 1 = small (up to the size of a fingertip for pileate or a few fingerprints for resupinate), 2 = intermediate (up to the size of an apple-half for pileate or a palm for resupinate) and 3 = large (larger than the size of a fist for pileate or two palms for resupinate). We added this information, together with the measurement of other traits, in the caption of Table S1, I which we list the species traits. Additional information on the measurement of fruit body traits can be obtained from Nordén et al. (2013) and Maurice et al. (2021).

It is indeed important to use quantitative measurement of fruit bodies when assessing intraspecific variation, as we have done in another study (Langeland et al., Fungal Ecology 2020). However, the fruit body size between species can be impacted by different factors (e.g. substrate volume, tree species, climate), and physical obstacles (branches, bark, rock, etc.). Therefore, from a technical and mycological perspective, it is less practical and coherent to quantitatively compare size across species, though we fully agree that classes can be defined on quantitative measurement.

Lines 122-123: Please include a citation for hyphal system complexity correlated to toughness.

>>> Included.

“The latter describes fruit body toughness where species with monomitic hyphae have softer fruit bodies, and species with dimitic and trimitic hyphal systems are progressively tougher [20, 21].” (line 123-125)

Lines 129-130 & 279-282: How were you able to collect interior tissue that was not contaminated from resupinate species? As you mention, for very thin species, being able to aseptically excise interior tissue to avoid contamination from transient species or grazers is critical to make conclusions about communities of arthropods hosted by these fungi. The weak but significant effect for high diversity in resupinate species is harder to accept due to this.

>>> We processed both annual and perennial species similarly, by cutting out the surface layer (indeed more delicate for resupinate species) and processed the subiculum layer. No tissue sterilisation was performed, as it would have been a source of bias in estimating diversity of fungicolous fungi (that we investigated in Maurice et al., 2021), which may have different colonisation patterns. Even though tissue is sterilised, DNA from dead fungal cells may remain and get unevenly amplified during the PCR (differing from culturing methods). Precisions have been brought in the manuscript:

“Briefly, we processed all fruit bodies similarly, by cutting out the outer surface layer, to avoid surface contaminants.” (line 130-131)

Lines 183-188: What is the justification for using 5 univariate models and information criteria instead of a multivariate model? Did you test for collinearity? A better approach might be to use a multivariate model for uncorrelated variables to acknowledge multiple factors that might be influencing the response. If unable to use a multivariate model because of collinearity, a measure of model fit should be included to show support for your chosen model.

>>> We used univariate models instead of a multivariate one mainly because of collinearity. From a fungal ecological point of view, many of the traits measured here are correlated. In addition, from preliminary tests it was clear that we were dealing with multicollinearity because the effect sizes and p values changed drastically when some variables were included in the same model. We have now added a measure of model fit for each univariate model (Marginal R^2) in Table 1, as suggested. Furthermore, we added some clarity in the Supplementary Material S8, including a table showing pair-wise Spearman’s rank correlation coefficient (ρ) between all variables and respective correlation tests.

Line 190: alpha set to 0.5 (shouldn’t this be 0.05?)

>>> Yes! Corrected.

“...and significance ($\alpha = 0.05$)...” (line 191)

Lines 268-270: Help support this section by reminding readers which of the fungal morphology types are homogeneous and heterogeneous.

>>> Corrected.

“Wood-decay fruit bodies, especially pileate, harbour several niches as they consist of different structural layers that vary in texture...” (line 263-264)

Line 314: Slightly confusing wording.

>>> Corrected.

“...as none were designed to amplify arthropods in forest ecosystem.” (line 302)

Line 322: Slightly confusing wording. Consider, “as hyphal systems can ALSO be correlated with other important factors, such as persistence or DIFFERING ability to hold water or nutrients”

>>> Thank you for your suggestion. We corrected to:

“Hyphal systems can also be correlated with other factors, such as persistence or differing ability to hold water or nutrients.” (line 323-324)

Tables: The commas in the numbers is confusing for me (although I acknowledge that it is more common in parts of Europe)

>>> Thank you for bringing this point to our attention, we brought modifications in all tables.

Figures:

Figure 1: To make this paper more accessible to a general audience, I suggest adding representative photographs/images of resupinate vs pileate species here. Clustering your species by type and not alphabetically (for panels 1a/c) would help a reader see patterns in the data.

>>> Concerning images to illustrate resupinate and pileate fruit bodies, we have included a Supplementary Figure S1, combined with drawings of the hyphal system (mono-, di- and trimitic) from Ryvar den 1989. In our manuscript, we made the choice of supporting our results with only three figures that we consider already full of information. As suggested, we have now reordered the species in Figure 1 according to resupinate/pileate morphologies.

Figure 2 legend, Line 614: How do these stress values relate to ones in “normal” NMDS?

>>> See answer above.

Figure 2: Showing your actual data points in the NMDS would be more informative.

>>> Yes, we agree this is informative. We have added this in Figure S6.3 in the Supplementary Material.

For either figure 2 or 3 it would be good to include an image depicting monomictic, dimictic and trimictic morphology

>>> See comment above regarding Figure 1.

Figure 4: It would be nice to see some specific examples highlighted here.

>>> Thank you for your suggestion, but we decided not to add more detail to the visualised network (Figure 3). However, we refer to specific examples in the main text, see lines 347-352. For more specific examples, the data underlying the co-occurrence network are available in Table S10.

Supplementary Figure 5: Your chosen rarefaction cutoff for this paper is fine, but I want to clarify that you do not want to choose a cutoff below the asymptote saturation. It is much better if you have enough sequencing depth to choose a cutoff after your alpha diversity has saturated. The text is a bit confusing here.

>>> Thank you for this comment. We have now clarified this in the Supplementary Material S5.

Referee: 2

Comments to the Author(s)

This is an interesting and novel study examining the communities of arthropods associated with the fruiting bodies of wood decay fungi. With the rise in the number of studies utilizing metabarcoding to investigate hidden diversity, it is surprising that the diversity within living fungi has been neglected in contrast to the extensive literature on plant-associated communities. This study begins to fill in that gap and points to exciting directions for future work. How analogous are fungus-arthropod interactions to the multifaceted ecological and evolutionary processes known from plant-arthropod interactions? This paper is an important first step. Not only do the authors reveal the hidden diversity of fungus-associated arthropods, but they also address some interesting more general questions, relating to the relationship between host specialization and ephemerality of the resource, how host traits influence their communities, and how habitat size influences community assembly. The counter-intuitive association between fruiting body size and arthropod diversity is noteworthy; unlike with larger ecosystems, host-associated symbiotic communities, to my knowledge, do not consistently follow the rules of island biogeography theory, so this is another example where the expectation from theory developed from larger organisms is not met. The level of host specificity in the OTUs observed is fascinating and opens up many questions as to the factors selecting for and maintaining these interactions. Finally, not only is this study an important contribution based on the particular subject matter, but I believe the methods are of note for metabarcoding approaches. Namely, a convincing case was made for the value of actually testing whether ASVs or OTUs are the more appropriate approach for the particular system in question. In short, the subject matter is original, the methods are sound and rigorous, and the results and conclusions are noteworthy. I mostly only have relatively minor suggestions:

>>> We thank you for the positive and helpful comments. Your suggestions have improved the accuracy of our manuscript. We provide below an answer to each of your concerns, including the ones on pileate/resupinate and ordered Figure 1 according to the fruit body morphologies.

Page 4 line 73: Do "predators" and "parasites" here refer to arthropods which predate on other arthropods and parasitoids, respectively? The use of "parasite" is particularly ambiguous as it could conceivably relate to the nature of the interaction between host fungus and symbiotic arthropod (as opposed to mutualistic). Using "parasitoid" at least would make clear the point that this is an ecosystem with multiple trophic levels beyond the direct fungus-arthropod interaction. Also, there's no discussion of the potential ecological roles of the observed OTUs later on. Are the patterns of diversity you observe driven by fungivores or are OTUs belonging to (the probably more transiently-associated) category of predators also important for driving certain trends. [see comment about the caveat on page 9 below, this is a related point]

>>> Thank you for putting this to our attention. We have now changed to the term "parasitoid". It is indeed a very interesting discussion and we have added details in the paragraph where we discuss ecological roles. However, we want to avoid further speculation on ecological roles based on our dataset, because of low resolution of OTU annotations to species level. Additional

DNA markers (or of longer length) are needed to discuss the potential ecological roles of OTUs with confidence.

“...predators or parasitoids.” (line 74)

“...and predators could make up to a third of the faunal composition in fruit bodies [25, 39]” (line 329)

Page 7 line 170: A strong methodological argument for using OTUs over ASVs in this case.

>>> Indeed. Thank you.

Page 7 line 173: It would be of interest to report what percentage of OTUs were annotated to fungi. Was it close to the remaining 54%?

>>> Arthropoda was by far the subphylum with most annotations. The second was Ochrophyta from 262 OTUs (7.2% of annotations). It is, however, a good point that the percentage of other annotations, especially of fungi, from a COI marker designed for arthropods is of interest to report. We have now added that.

“The remaining OTUs were annotated to Ochrophyta (7.2%), Vertebrata (5%), Basidiomycota (4.5%), Mollusca (4.2%) and Ascomycota (3.9%), among others.” (line 176-177)

Page 9 line 222: I find it quite surprising that a fifth of the sequences could not be identified below the level of Arthropoda. Is this a consequence of a limited reference database (does it not contain all known orders of arthropods?). Alternatively, do you suspect there's a major risk of degradation prior to extraction—might enzymes inside the fungal fruiting bodies decrease the integrity of the DNA within the sample?

>>> The main reason behind this low taxonomic resolution, can be that Arthropoda contains many more orders not referenced neither in the BOLD nor NCBI databases, despite them being the most complete ones for COI. Our study on Arthropoda within several fungal species touched a new habitat, until now poorly studied with metabarcoding, which could explain why we find low correspondence with the databases.

However, as we were able to amplify, sequence, and obtain a blast hit to Arthropoda for the 412 bp region targeted in our study (COI markers BR2/BF3), the quality of the DNA is probably not the limiting factor, but instead a limit of the database.

Page 9 lines 279-282: Could this caveat be addressed by comparing the results of metabarcoding the outer vs. inner layers of pileate fruit bodies? Presumably, the risk of 'contamination' from transient visitors should be more apparent on the outer surfaces, especially if there are taxa that are seen on resupinate fungi and the outer surfaces of pileate ones, but not in the interiors of the pileate fungi. Also, if such contamination is at play it should be apparent from examining the particular taxonomic composition of your resupinate samples. Based on the most frequent families you report, it appears that the major taxa you've observed are mostly arthropods which are known to dwell in fungi. It would be quite telling if the OTUs driving the higher diversity in resupinate samples are annotated as less clearly fungus-associated taxa. Can you do an ANCOM test to determine which are the OTUs that occur only in your resupinate samples?

>>> We do not have samples from outer and inner layers of fruit bodies (inner only), but it could indeed be a way to address this issue.

As suggested, we investigated further the OTUs underlying the diversity patterns between the two fruit body morphologies. We selected the most common OTUs (top 20) and looked at

their relative distribution in both resupinate and pileate species. Given the low taxonomic resolution beyond order-level, these results were not very informative, and were therefore not included in the manuscript. For example, the third and fifth most common OTUs (OTU0851 and 0854, respectively) are annotated to the family Bolitophilidae (Diptera). OTU0854 was common in resupinate fruit bodies, whereas OTU0851 was not – but only around half of bolitophilid species are known to develop in fungal fruit bodies (Sevcík 2010). Not knowing which species are present in our dataset, any inference based on ecological guilds would be highly uncertain and critical from a taxonomic point of view.

Yet, some positive assurance is given against contamination because the patterns we show are strongly supported by the statistics. All three multivariate tests were unambiguous in finding significant differences in arthropod composition between resupinate and pileate fruit bodies. Furthermore, the indicator species analysis, which looked for consistent patterns across individuals of each fungal species, also identified co-occurrences between resupinate species and OTUs from fungus-dwelling genera/families.

Page 10 line 290: This will be an especially interesting topic of future investigation, along with just generally investigating the role of physiological characteristics of the living fungal hosts.

>>> Indeed, that would be very interesting

Figure 1: Is there any way to indicate which species are pileate and which are resupinate on this figure? Perhaps you can design small representative icons to place right next to the abbreviated species names. You'd also be able to easily fit the key directly above or below the taxonomic key for the stacked barplot. Or maybe even more simply, just change the color coding for the diversity barplots in (1a) to match the color code for (1b). As it stands, assigning each individual species a separate color does not add much information (other than creating continuity with Fig. 2, but the key in that figure suffices). It would carry more information to have open boxes for pileate species and solid grey for resupinate in (1a).

>>> We reorganised the host by these two levels, and included a legend. We hope the message is clearer.

References:

Jenssen, Andreas Langeland, Håvard Kauserud, and Sundy Maurice. "High phenotypic variability in the wood decay fungus *Phellopilus nigrolimitatus*." *Fungal Ecology* (2020): 100982.

Maurice, Sundy, et al. "Fungal sporocarps house diverse and host-specific communities of fungicolous fungi." *The ISME journal* 15.5 (2021): 1445-1457.

Norden, Jenni, et al. "Specialist species of wood-inhabiting fungi struggle while generalists thrive in fragmented boreal forests." *Journal of Ecology* 101.3 (2013): 701-712.

Ševčík, Jan. *Czech and Slovak Diptera associated with fungi*. Opava: Slezské zemské muzeum, 2010.

Appendix B

Associate Editor

Board Member

Comments to Author:

I agree with the reviewer that the suggested revisions have been made to a high standard. A very small number of minor issues remain, which should be addressed.

>>> Dear Editor,

thank you for your response. We have addressed the minor issues as pointed out by the referee and yourself by (1) moving the Data Accessibility section and including the Zenodo accession number, and (2) changing all “gNMDS” to “NMDS” in the text and responding to the other concerns below.

Reviewer(s)' Comments to Author:

Referee: 1

Comments to the Author(s).

As noted in my previous review, I believe this paper presents interesting findings that add to our ecological knowledge of fungal host traits and their effect on the diversity and structure of associated arthropod communities. I value the combination of traditional mycological approaches using morphology coupled with diversity metrics using sequencing data.

The authors have addressed my main concerns by: correcting the stress values for the NMDS and adding more information about the approach, clarifying why they chose to use univariate models, adding photographs in the supplementary material, and making the data more accessible to reviewers and readers. They also re-organized the species in Figure 1 to show the results more clearly.

The authors should also specify that the R code is deposited in Zenodo and include the link in the text under “Data accessibility” or just include the R code in the Dryad folder.

>>> Corrected.

I'm confused as to why the authors chose to use a global version of monoMDS (and then doing a Procrustes analysis) instead of using metaMDS, which does all this for you while also using several random starts. Either way, I find the “gNMDS” designation misleading because, as the authors confirm in their response, they are using what is considered the normal form of NMDS. I think it is OK to state that it is a global NMDS when first introduced, but all the “gNMDS” should be changed to simple “NMDS” to align with norms and other publications. If the analysis is indeed different in some way than it would be with metaMDS, then the authors should justify their approach and state how it is different.

>>> Corrected all 'gNMDS' to 'NMDS' in the text.

We used the monoMDS function because it gives more control over the ordination process. As metaMDS calls for the function monoMDS to generate the NMDS, the difference seems to be mostly regarding the user's preference. Mainly, metaMDS saves text in the script and seems to be more user-friendly.

However, we prefer the more 'tedious' and formerly used approach because it not only provides a better overview of the different steps, but it can also be readily modified by changing settings and options that may largely influence the ordination process, such as: choice of NMDS variant, convergence criteria (sfgmin and smin arguments), maximum number of iterations, *a posteriori* scaling and rotation (with the postMDS function) and criteria for selecting among alternative NMDS configurations. For more rigour in the ordination analysis, we would encourage more researchers to do it this way.

Overall, I appreciate the correction of typos and terminology.